# *In silico* approaches for drug repurposing in oncology: Protocol for a scoping review of existing evidence

**Bruno Raphael Ribeiro Cavalcante**[1,2☯], **Raíza Dias Freitas**[3☯], **Leonardo de Oliveira Siquara da Rocha**[1,2], **Gisele Vieira Rocha**[1,4], **Túlio Cosme de Carvalho Pachêco**[5], **Pablo Ivan Pereira Ramos**[1,6], **Clarissa Araújo Gurgel Rocha**[1,2,4,7] *

1 Gonçalo Moniz Institute, Oswaldo Cruz Foundation (IGM-FIOCRUZ/BA), Salvador, Brazil, 2 Department of Pathology and Forensic Medicine, School of Medicine, Federal University of Bahia, Salvador, Brazil, 3 School of Dentistry, University of São Paulo, São Paulo, Brazil, 4 D'Or Institute for Research and Education (IDOR), Salvador, Brazil, 5 Bahiana School of Medicine and Public Health, Salvador, Brazil, 6 Center of Data and Knowledge Integration for Health (CIDACS), Salvador, Brazil, 7 Department of Propaedeutics, School of Dentistry of the Federal University of Bahia, Bahia, Brazil

☯ These authors contributed equally to this work.
* clarissa.gurgel@fiocruz.br, gurgel.clarissa@gmail.com

**Data Availability Statement:** No datasets were generated or analysed during the current study. All

## Abstract

Drug repurposing has been applied in the biomedical field to optimize the use of existing drugs, leading to a more efficient allocation of research resources. In oncology, this approach is particularly interesting, considering the high cost related to the discovery of new drugs with therapeutic potential. Computational methods have been applied to predict associations between drugs and their targets. However, drug repurposing has not always been promising and its efficiency has yet to be proven. Therefore, the present scoping review protocol was developed to screen the literature on how *in silico* strategies can be implemented in drug repurposing in oncology. The scoping review will be conducted according to the Arksey and O'Malley framework (2005) and the Joanna Briggs Institute recommendations. We will search the PubMed/MEDLINE, Embase, Scopus, and Web of Science databases, as well as the grey literature. We will include peer-reviewed research articles involving *in silico* strategies applied to drug repurposing in oncology, published between January 1, 2003, and December 31, 2021. Data will be charted and findings described according to review questions. We will report the scoping review using the Preferred Reporting Items for Systematic Reviews and Meta-Analyses Extension for Scoping Review guidelines (PRISMA-ScR).

## Introduction

A variety of diseases have been part of human history, and although many of them have already been eradicated or have effective treatment, we still face the persistence of those that are still currently present. Medicines played a pivotal role in this process in which drug formulations were used to refrain disease progression at a molecular and physiological level. In this context, cancer, a set of diseases defined by an uncontrolled abnormal cell growth, defies new

relevant data from this study will be made available upon study completion.

**Funding:** This work was financially supported by the following institutions: (1) National Support Program for Oncology Care (PRONON) [#25000.192875/2019-78], (2) Conselho Nacional de Desenvolvimento Científico e Tecnológico (CNPq) [#308276/2019-1], (3) INOVA Fiocruz [PP1192029]. BRRC is supported by the Gonçalo Moniz Institute, Oswaldo Cruz Foundation. RDF is funded by Fundação de Amparo à Pesquisa do Estado de São Paulo [#2018/03199-6]. LSR is supported by CNPq [#9196892774005300]. GVR is funded by Maria Emília Foundation (Salvador, Bahia, Brazil). CAGR is granted for research productivity with a scholarship award by CNPq. The funders had and will not have a role in study design, data collection and analysis, decision to publish, or preparation of the manuscript.

**Competing interests:** The authors have declared that no competing interests exist.

medicine and keeps demanding action as the lack of approved drugs continues to be a challenge and further requires the discovery of new therapeutic molecules.

It is noteworthy that the cost of new drug development has been increasing over the years. Indeed, the process of medicinal formulation from invention to the pharmacy shelves ranged from $1.1 billion in 2003 to $2.8 billion in 2013 [1]. Also, this is a time-consuming and high-risk process, as the research and development take about 10–15 years for a single new drug to be commercially available, with a success rate of developing a new molecular entity being only 2.01% [2, 3]. This current scenario has forced drug developers to become more innovative in finding new therapeutic applications for existing drugs that are outside the scope of their original medical indication, a strategy known as drug repurposing (also called drug repositioning, reprofiling or re-tasking) [4]. Therefore, repurposing existing drugs for new therapeutic assignments is considered a better approach and serves as an important way to hasten drug discovery, especially in cancer therapy.

Identifying alternative purposes for known drugs is important for the pharmaceutical industry as well as for patients. However, this is not an easy journey: exploring the large collection of data commonly seen in the Big Data era requires additional efforts related to data extraction and, most importantly, data analyses. In this sense, the emergence of Computational methods has revolutionized life science studies, especially in terms of repositioning drugs, as they aim to identify the relationship network between target and drugs. In fact, many computational methods, including artificial intelligence techniques, have been proposed for predicting unknown associations between drugs and target proteins associated with diseases, in an attempt to tackle the traditional limitations of bulky information [3]. Succinctly, in silico approaches help in screening large compound libraries at once, can be applied to drug-target and toxicity predictions, and, most importantly, relieve the pressure concerning the costs of laboratory works and animal sacrifices [5].

In practical terms, the gap between what can be found through computational strategies and the lacking knowledge concerning drug repurposing in oncology is yet to be completely demonstrated. Although many reports show promising results, application of these strategies in drug repurposing does not always succeed, indicating a possible misleading and failed use in several cases so far, mostly due to the questionable chosen input data, poor data quality and debatable analysis methods. Besides, patent considerations, regulatory concerns and organizational hurdles contribute to delay to-be-repurposed drugs from reaching clinical practice [6]. Hence, we designed the present scoping review protocol to unravel how the *in silico* strategies can be implemented in drug repurposing in oncology, surpassing all current caveats.

## Materials and methods

### Study design and registration

This scoping review protocol was developed according to the framework proposed by Arksey and O'Malley [7], and complies with the recommendations of the Joanna Briggs Institute for elaborating scoping reviews [8]. The protocol is reported according to the Preferred Reporting Items for Systematic review and Meta-Analysis Protocols (PRISMA-P) [9] (S1 Checklist). Moreover, it was registered with Open Science Framework (osf.io/yx7kp).

### Review questions

Considering the review aim, we intend to answer the following research questions:

1. How can *in silico* strategies be implemented in drug repurposing in the oncology field?

2. Which *in silico* strategies are mostly used for drug repurposing in oncology?

3. What are the regulatory barriers to drug repurposing when using *in silico* strategies in cancer research?

## Search strategy

The PubMed/Medline search strategy was developed and adapted for the other databases (Embase, Scopus, and Web of Science). Furthermore, we will search for grey literature in Grey Literature in Europe (Open Grey). Our search strategy combines Medical Subject Headings (MeSH) terms, their relevant synonyms, and the Boolean operators "AND" and "OR". Three concept clusters were included: (1) *in silico* approach, (2) drug repurposing, and (3) oncology. The description of the search strategies which will be used in each database can be found in Table 1.

We will perform the search on each database and export the results in Comma-Separated Values (CSV) format to Microsoft Excel. Subsequently, duplicates will be removed using their PubMed ID and title, and reviewers will proceed with the inclusion and exclusion of the studies.

## Study selection

**Inclusion criteria.** In addition to the parameters of the search strategy "Table 1", studies will be included if they meet the following criterion:

- Peer-reviewed research articles, published between January 1, 2003, and December 31, 2021

- Studies that have implemented *in silico* strategies for drug repurposing in oncology

**Exclusion criteria.** Studies will be excluded if they meet the following criterion:

- Narrative or systematic reviews, book chapters, author's opinion/comments, editorials, erratum, meeting abstracts, conference abstracts and study protocols

**Table 1. Search strategies according to the electronic databases.**

| Database | Search Strategy |
|---|---|
| PubMed | (("drug repositioning" OR "drug repurposing" OR "drug rescue" OR "off-label use" OR "off-label uses" OR "off-label prescribing" OR "unlabeled indication" OR "high throughput screening" OR "high throughput screening assays") AND ("*in silico*" OR "*in silico*s" OR "computer simulation" OR "computerized model")) AND (oncolog* OR cancer* OR tumor* OR tumour* OR neoplas*)) |
| Embase | ('drug repositioning'/exp OR 'drug repurposing'/exp OR 'drug rescue'/exp OR 'off-label use' OR 'off-label uses' OR 'off-label prescribing'/exp OR 'unlabeled indication' OR 'high throughput screening'/exp OR 'high throughput screening assays'/exp) AND ("*in silico*" OR "*in silico*s" OR "computer simulation" OR "computerized model") AND (oncolog* OR cancer* OR tumor* OR tumour* OR neoplas*) |
| Scopus | TITLE-ABS-KEY (("drug repositioning" OR "drug repurposing" OR "drug rescue" OR "off-label use" OR "off-label uses" OR "off-label prescribing" OR "unlabeled indication" OR "high throughput screening" OR "high throughput screening assays") AND ("*in silico*" OR "*in silico*s" OR "computer simulation" OR "computerized model") AND (oncolog* OR cancer* OR tumor* OR tumour* OR neoplas*)) |
| Web of Science | TS = (("drug repositioning" OR "drug repurposing" OR "drug rescue" OR "off-label use" OR "off-label uses" OR "off-label prescribing" OR "unlabeled indication" OR "high throughput screening" OR "high throughput screening assays") AND ("*in silico*" OR "*in silico*s" OR "computer simulation" OR "computerized model") AND (oncolog* OR cancer* OR tumor* OR tumour* OR neoplas*)) |
| Open Grey | (("drug repositioning" OR "drug repurposing" OR "drug rescue" OR "off-label use" OR "off-label uses" OR "off-label prescribing" OR "unlabeled indication" OR "high throughput screening"OR "high throughput screening assays") AND ("*in silico*" OR "*in silico*s" OR "computer simulation" OR "computerized model") AND (oncolog* OR cancer* OR tumor* OR tumour* OR neoplas*)) |

- Studies that used *in silico* strategies for other objectives rather than drug repurposing

- Studies where no abstract is available or where full-text articles cannot be obtained

- Studies not written in English

- Studies on drug repurposing in oncology intended for animal use

Two independent reviewers (B.R.R.C. and L.O.S.R.) will perform the studies' screening and selection and inter-rater agreement will be assessed through Cohen's κ at the abstract review stage. The reviewers are PhD students from the Pathology postgraduate program at the Gonçalo Moniz Institute (Oswaldo Cruz Foundation) who have been involved in research applying in silico methods in oncology. All disagreements will be discussed with a third reviewer with deep experience on *in silico* methods. Firstly, the reviewers will evaluate the studies' titles and abstracts against the inclusion criteria, indicating the relevant studies to be included. Subsequently, the reviewers will assess the studies through their full-text according to the exclusion criteria.

## Data charting

Data charting will be a descriptive summary of the results, and qualitative data synthesis will be performed according to our review questions. Data charting will be piloted by two reviewers and adjustments will be made as required. We will extract the following data: (1) title, (2) date of the publication, (3) authors, (4) country, (5) study aim, (6) study design, (7) type of cancer, (8) *in silico* method implemented, (9) study outcome, (10) whether regulatory aspects are mentioned, (11) whether drug rescue or repurposing was performed. The *in silico* method implemented in the study will be charted as described by the authors. Data charting will be performed and stored using Microsoft Excel by the reviewers. Prior to data extraction, the spreadsheet containing 11 columns with the abovementioned topics will be tested on 10 randomly selected studies and corrections will be made whether needed. Disagreements between the reviewers will be amended by a third reviewer. Authors of the included papers will be contacted in case of missing information.

## Data summary

We intend to present our results in a narrative form and with the use of tables containing the topics detailed in the previous section. Findings will be described according to the review questions objectively and the results section content may be further adjusted after reviewing the studies. We will use the PRISMA-ScR checklist to report the scoping review.

## Changes to the protocol

Changes to the study protocol will only be made if needed and will be reported accordingly.

## Discussion

*In silico* methods have been widely used for drug repurposing in the biomedical field. However, their significance, efficiency and merit have yet to be fully appraised. In order to overcome the challenges in the research of drug repositioning, the results from this scoping review will be the first step on mapping how successful these methods have been since most of them are hypothesis-driven approaches that takes advantage of the use of Big data. This will be achieved by showing evidence on which strategies have reached a clinical application or are prone to be used in oncology research. Additionally, to circumvent the issues related to the

efficacy/effectiveness and safety of drugs, the integration of multi-source information regarding drugs and their side effects and interactions of drugs and cancer will be addressed.

We intend to provide valuable information for researchers from diverse areas in understanding the favourable *in silico* methods used in oncology, and how they may be incorporated alone or in association with other methods in the process of drug repositioning. Thus, we aim to contribute to the future application of the *in silico* methods in drug repurposing based on the best evidence available.

## Supporting information

**S1 Checklist. Preferred Reporting Items for Systematic review and Meta-Analysis Protocols (PRISMA-P) checklist.**
(PDF)

## Author Contributions

**Conceptualization:** Bruno Raphael Ribeiro Cavalcante, Clarissa Araújo Gurgel Rocha.

**Formal analysis:** Bruno Raphael Ribeiro Cavalcante, Raíza Dias Freitas, Leonardo de Oliveira Siquara da Rocha, Clarissa Araújo Gurgel Rocha.

**Funding acquisition:** Clarissa Araújo Gurgel Rocha.

**Investigation:** Bruno Raphael Ribeiro Cavalcante, Raíza Dias Freitas, Leonardo de Oliveira Siquara da Rocha.

**Methodology:** Bruno Raphael Ribeiro Cavalcante, Raíza Dias Freitas, Leonardo de Oliveira Siquara da Rocha, Gisele Vieira Rocha, Túlio Cosme de Carvalho Pachêco, Pablo Ivan Pereira Ramos, Clarissa Araújo Gurgel Rocha.

**Project administration:** Clarissa Araújo Gurgel Rocha.

**Supervision:** Pablo Ivan Pereira Ramos, Clarissa Araújo Gurgel Rocha.

**Writing – original draft:** Bruno Raphael Ribeiro Cavalcante, Raíza Dias Freitas, Leonardo de Oliveira Siquara da Rocha.

**Writing – review & editing:** Bruno Raphael Ribeiro Cavalcante, Raíza Dias Freitas, Leonardo de Oliveira Siquara da Rocha, Clarissa Araújo Gurgel Rocha.

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
