## [Decision Letter · Decision Letter 0]

24 Feb 2022

PONE-D-21-38613IN SILICO APPROACHES FOR DRUG REPURPOSING IN ONCOLOGY: PROTOCOL FOR A SCOPING REVIEW OF EXISTING EVIDENCEPLOS ONE

Dear Dr. Gurgel Rocha,

Thank you for submitting your manuscript to PLOS ONE. After careful consideration, we feel that it has merit but does not fully meet PLOS ONE’s publication criteria as it currently stands. Therefore, we invite you to submit a revised version of the manuscript that addresses the points raised during the review process.

We look forward to receiving your revised manuscript.

Kind regards,

Vivek Gupta

Academic Editor

PLOS ONE

Journal Requirements:

Reviewers' comments:

Reviewer's Responses to Questions

**Comments to the Author**

1. Does the manuscript provide a valid rationale for the proposed study, with clearly identified and justified research questions?

Reviewer #1: Yes

Reviewer #2: Yes

2. Is the protocol technically sound and planned in a manner that will lead to a meaningful outcome and allow testing the stated hypotheses?

Reviewer #1: Yes

Reviewer #2: Yes

3. Is the methodology feasible and described in sufficient detail to allow the work to be replicable?

Reviewer #1: Yes

Reviewer #2: Yes

4. Have the authors described where all data underlying the findings will be made available when the study is complete?

Reviewer #1: Yes

Reviewer #2: Yes

5. Is the manuscript presented in an intelligible fashion and written in standard English?

Reviewer #1: Yes

Reviewer #2: Yes

6. Review Comments to the Author

You may also provide optional suggestions and comments to authors that they might find helpful in planning their study.

Reviewer #1: 1. Authors are recommended to include few statements on the importance of in silico approaches in the introduction. (justifying the reason to develop the current protocol)

2. Discussion section needs to be elaborated with more details on the current challenges towards drug repurposing in oncology and how the protocol could address them.

3. Data Charting: Data (8): in silico method implemented – Please mention if this refers to a method alone or methods in combination.

4. Data Charting: This could be explained for a clear understanding.

5. Line 46: Please check for the sentence completion and modify as needed.

6. Is there any specific criteria in reviewers' selection?

Reviewer #2: Authors proposed to submit a protocol to study the published literature about drug re-purposing. Drug re-purposing is might be more feasible way to develop new and effective treatment strategies for diseases like cancer. Authors did a great job in explaining search key words and tools to collect published data in this field.

7. PLOS authors have the option to publish the peer review history of their article (what does this mean?). If published, this will include your full peer review and any attached files.

Reviewer #1: **Yes: **Vineela Parvathaneni

Reviewer #2: **Yes: **Ashwni Verma

---

## [Author Response · Author response to Decision Letter 0]

18 Apr 2022

We would like to thank the reviewers for their comments regarding our study. The suggested amendments significantly improved the quality and readability of the manuscript. 

Reviewer #1: 

1. Authors are recommended to include few statements on the importance of in silico approaches in the introduction. (justifying the reason to develop the current protocol).

A:Thank you for your comment. We have included a few statements to support the purpose of this study (lines 68-80), as follows:

“Identifying alternative purposes for known drugs is important for the pharmaceutical industry as well as for patients. However, this is not an easy journey: exploring the large collection of data commonly seen in the Big Data era requires additional efforts related to data extraction and, most importantly, data analyses. In this sense, the emergence of Computational methods has revolutionized life science studies, especially in terms of repositioning drugs, as they aim to identify the relationship network between target and drugs. In fact, many computational methods, including artificial intelligence techniques, have been proposed for predicting unknown associations between drugs and target proteins associated with diseases, in an attempt to tackle the traditional limitations of bulky information [3]. Succinctly, in silico approaches help in screening large compound libraries at once, can be applied to drug-target and toxicity predictions, and, most importantly, relieve the pressure concerning the costs of laboratory works and animal sacrifices [5].”

2. Discussion section needs to be elaborated with more details on the current challenges towards drug repurposing in oncology and how the protocol could address them.

A: We agree with your comment. We have made amendments to the discussion section comprising the current challenges and how we intend to aid in this field (lines 180-188), as follows:

“In order to overcome the challenges in the research of drug repositioning, the results from this scoping review will be the first step on mapping how successful these methods have been since most of them are hypothesis-driven approaches that takes advantage of the use of Big data. This will be achieved by showing evidence on which strategies have reached a clinical application or are prone to be used in oncology research. Additionally, to circumvent the issues related to the efficacy/effectiveness and safety of drugs, the integration of multi-source information regarding drugs and their side effects and interactions of drugs and cancer will be addressed.”

3. Data Charting: Data (8): in silico method implemented – Please mention if this refers to a method alone or methods in combination.

A:Thank you for your comment. The method implemented will be charted as described by the authors of the included study. Therefore, both possibilities will be considered. This information was described in lines 162 and 163. 

“The in silico method implemented in the study will be charted as described by the authors.”

4. Data Charting: This could be explained for a clear understanding.

A:We have added more details to improve the understanding of data charting (lines 164-168).

“Prior to data extraction, the spreadsheet containing 11 columns with the abovementioned topics will be tested on 10 randomly selected studies and corrections will be made whether needed. Disagreements between the reviewers will be amended by a third reviewer. Authors of the included papers will be contacted in case of missing information.”

5. Line 46: Please check for the sentence completion and modify as needed.

A: Thank you. This sentence was excluded from the abstract. 

6. Is there any specific criteria in reviewers' selection?

A:The reviewers are PhD students from our Pathology program who have been involved in applying in silico methods in oncology. We have added this information on lines 145-149, as follows:

“Two independent reviewers (B.R.R.C. and L.O.S.R.) will perform the studies’ screening and selection and inter-rater agreement will be assessed through Cohen’s κ at the abstract review stage. The reviewers are PhD students from the Pathology postgraduate program at the Gonçalo Moniz Institute (Oswaldo Cruz Foundation) who have been involved in research applying in silico methods in oncology.”

Reviewer #2: Authors proposed to submit a protocol to study the published literature about drug re-purposing. Drug re-purposing is might be more feasible way to develop new and effective treatment strategies for diseases like cancer. Authors did a great job in explaining search key words and tools to collect published data in this field.

A:Thank you for your comment on our work. This understanding is as intended by the authors.

---

## [Decision Letter · Decision Letter 1]

22 Jun 2022

In silico approaches for drug repurposing in oncology: protocol for a scoping review of existing evidence

PONE-D-21-38613R1

Dear Dr. Gurgel Rocha,

We’re pleased to inform you that your manuscript has been judged scientifically suitable for publication and will be formally accepted for publication once it meets all outstanding technical requirements.

Kind regards,

Vivek Gupta

Academic Editor

PLOS ONE

Additional Editor Comments (optional):

Reviewers' comments:

Reviewer's Responses to Questions

**Comments to the Author**

1. Does the manuscript provide a valid rationale for the proposed study, with clearly identified and justified research questions?

Reviewer #1: Yes

2. Is the protocol technically sound and planned in a manner that will lead to a meaningful outcome and allow testing the stated hypotheses?

Reviewer #1: Yes

3. Is the methodology feasible and described in sufficient detail to allow the work to be replicable?

Reviewer #1: Yes

4. Have the authors described where all data underlying the findings will be made available when the study is complete?

Reviewer #1: Yes

5. Is the manuscript presented in an intelligible fashion and written in standard English?

Reviewer #1: Yes

6. Review Comments to the Author

You may also provide optional suggestions and comments to authors that they might find helpful in planning their study.

Reviewer #1: Authors have addressed all the comments carefully and made changes to the manuscript as needed. Thank you.

7. PLOS authors have the option to publish the peer review history of their article (what does this mean?). If published, this will include your full peer review and any attached files.

Reviewer #1: **Yes: **Vineela Parvathaneni

---

## [Editor Report · Acceptance letter]

27 Jun 2022

PONE-D-21-38613R1 

*In silico* approaches for drug repurposing in oncology: protocol for a scoping review of existing evidence 

Dear Dr. Gurgel Rocha:

I'm pleased to inform you that your manuscript has been deemed suitable for publication in PLOS ONE. Congratulations! Your manuscript is now with our production department. 

Kind regards, 

on behalf of

Dr. Vivek Gupta 

Academic Editor

PLOS ONE